# Presence of *Trypanosoma cruzi* (TcI) in different tissues of *Didelphis virginiana* from the metropolitan area of Merida, southeastern Mexico: Epidemiological relevance and implications for non-vector transmission routes

**Pedro Pablo Martínez-Vega**[1] *, Marian Rivera-Pérez[1,2], Gabrielle Pellegrin[1,3], Antoine Amblard-Rambert[1,3], Jorge Andrés Calderón-Quintal[1], Christian Barnabé[3], Christian Teh-Poot[1], Hugo Ruiz-Piña[4], Antonio Ortega-Pacheco[5], **Etienne Waleckx**[1,3,6] *

**1** Laboratorio de Parasitología, Centro de Investigaciones Regionales Dr. Hideyo Noguchi, Universidad Autónoma de Yucatán, Mérida, México, **2** Universidad Politécnica de Huatusco, Huatusco, México, **3** Institut de Recherche pour le Développement, UMR INTERTRYP IRD, CIRAD, Université de Montpellier, Montpellier, France, **4** Laboratorio de Zoonosis, Centro de Investigaciones Regionales Dr. Hideyo Noguchi, Universidad Autónoma de Yucatán, Mérida, México, **5** Departamento de Salud Animal y Medicina Preventiva, Facultad de Medicina Veterinaria y Zootecnia, Universidad Autónoma de Yucatán, Mérida, México, **6** ACCyC, Asociación Chagas con Ciencia y Conocimiento, A. C., Orizaba, Veracruz, México

* pedro.martinez@correo.uady.mx (PPMV); etienne.waleckx@ird.fr, etienne.waleckx@correo.uady.mx (EW)

## Abstract

### Background

*Trypanosoma cruzi* is mainly transmitted to mammals by vectors, but other transmission routes exist. For example, opossums can harbor the infectious form of the parasite in their anal glands, underscoring their potential role in non-vectorial transmission. *T. cruzi* has been detected in the anal gland secretions of various opossum species, and their infectivity has been confirmed in *Didelphis marsupialis* and *D. albiventris*. Vertical transmission has also been proposed in *D. virginiana*. However, if this occurs in opossums, it remains unclear whether it happens during pregnancy or lactation. In Mexico, *Didelphis virginiana* and *D. marsupialis* are the main opossum species. Our objective was to investigate the possible contribution of urban opossums to non-vectorial transmission of *T. cruzi* in the metropolitan area of Merida, Yucatan, in southeastern Mexico.

### Methodology/Principal findings

Blood, anal gland secretions, and milk were collected from opossums captured in Merida, Mexico, all identified as *D. virginiana* using taxonomic keys and Cytb sequencing. By PCR, *T. cruzi* was detected in 16/102 opossums (15.69%) in at least one type of sample. The prevalence was 14.71% (15/102) in blood and 0.98% (1/102) in anal gland secretions. 1/22 milk samples (4.55%) tested positive. Blood of 37 offspring from *T. cruzi*-positive mothers

**Data Availability Statement:** All data generated and analyzed during this study are included in the article and its additional files. Moreover, generated sequences were submitted to GenBank under accession numbers PQ516272-516278.

**Funding:** The author(s) received no specific funding for this work.

**Competing interests:** The authors declare that they have no competing interests.

was collected and tested negative. qPCR revealed that females with offspring tended to have lower parasite load in blood compared to females without offspring and males. Genotyping of the parasite through multiplex PCR revealed only the DTU TcI.

## Conclusions/Significance

This study agrees with previous works where *D. virginiana* was the most abundant opossum species in urban areas in southeastern Mexico and confirms that it is associated with TcI. Detection of *T. cruzi* in a sample of anal gland secretions underscores the potential risk represented by *D. virginiana* in non-vectorial transmission in urban areas of southeastern Mexico. Detection in the milk of a lactating female, along with the observed tendency towards a lower parasite load in females with offspring, highlight the importance of further investigating vertical transmission in *D. virginiana*.

### Author summary

Opossums are primary reservoirs of *T. cruzi*, the parasite responsible for Chagas disease, mainly transmitted to mammals by triatomine insects. Besides the vectorial route, opossums, which may have the unique ability among mammals to harbor the infective form of the parasite in their anal glands, may be involved in alternative transmission routes. Additionally, recent findings suggest that Virginia opossums (*Didelphis virginiana*) may vertically transmit *T. cruzi*, potentially increasing parasite prevalence within their populations and transmission risk via vectorial and non-vectorial routes. This makes opossums critical for studying *T. cruzi* transmission mechanisms, especially non-vector pathways. While *T. cruzi* has been detected in anal glands of various opossum species, transmission through gland secretions has only been confirmed in *Didelphis marsupialis* and *D. albiventris*. Moreover, if vertical transmission occurs in opossums, the mechanism remains unclear. We assessed the presence of *T. cruzi* in blood, anal gland secretions, and milk sampled in urban opossums from the metropolitan area of Merida, Yucatan, in southeastern Mexico. Detection of the parasite in anal gland secretions and milk suggests possible non-vectorial transmission routes. Absence of infection in offspring from infected opossum mothers, along with the observed tendency towards a lower parasite load in females with offspring, suggests that parasite load may constrain vertical transmission.

## Introduction

The protozoan parasite *T. cruzi* is the causative agent of Chagas disease, a complex zoonosis that is potentially fatal. It is estimated that between 6 and 7 million people worldwide are infected with *T. cruzi* [1]. The parasite is mainly transmitted through the feces/urine of hematophagous insects of the Triatominae subfamily contaminated with metacyclic trypomastigote forms of the parasite. However, transmission can also occur through non-vectorial routes, including but not limited to, blood transfusion, congenital transmission, organ transplantation, consumption of food and beverages contaminated with *T. cruzi* [1].

The life cycle of *T. cruzi* is complex, involving both vertebrate and invertebrate hosts and progressing through three distinct developmental stages. Amastigote forms are the proliferative stages that reside within the cells of the vertebrate host, while epimastigotes proliferate

within the intestine of the invertebrate host, giving rise to highly infectious metacyclic trypo-mastigotes [2].

*T. cruzi* exhibits a high genetic diversity, generally classified into seven discrete typing units (DTUs) named TcI to TcVI and Tcbat [3–5]. This diversity has been associated with different transmission cycles [5]. On the other hand, domestic and wild animals are essential reservoirs for maintaining the circulation of *T. cruzi*, as they sustain parasite populations and also serve as blood-feeding sources for triatomines [6]. Although more than 150 species of wild and domestic mammals have been described as hosts for *T. cruzi* [7], American marsupials of the genus *Didelphis* are recognized among the most important reservoirs [6,8,9]. In Mexico, the two most abundant species of opossums are *Didelphis marsupialis* and *D. virginiana*. A prevalence of infection with *T. cruzi* of up to 55% has previously been reported in these species in the country [10–14]. Due to their omnivorous diet, which may include insects infected with *T. cruzi* resulting in oral infection, and their synanthropic behavior closely associated with human habitats, opossums play a critical role in the transmission cycles of this parasite [15–17]. Furthermore, opossums may possess the unique ability among mammals to concurrently support both multiplication cycles of *T. cruzi* in their anal glands, simultaneously hosting both amastigotes and epimastigotes, as well as the highly infective metacyclic trypomastigotes [8]. Because opossums use to expel the content of their anal glands in response to threats or stress [18,19], this ability underscores their potential significance in the non-vectorial transmission of *T. cruzi*, which has been supported by successful infections of mice and opossums with anal gland secretions harboring the parasite [20,21].

Nevertheless, reports on the presence of *T. cruzi* in the anal glands are scarce, and the factors that define colonization of these glands are still unclear. It has been suggested that parasitemia levels may directly correlate with gland colonization [22]. Additionally, factors such as the opossum species and the specific strain of *T. cruzi* can influence this process. Anal gland infection has been confirmed in several opossum species [8,21,23,24], but the prevalence varies widely, ranging from 0 to 18.8% [21,24–27].

Furthermore, while *T. cruzi* has been identified in the anal gland secretions of *D. virginiana* [24,27], infectivity through these secretions has only been demonstrated in *D. marsupialis* and *D. albiventris* [20,21]. Additionally, based on the identification of *T. cruzi* DNA in offspring born from positive *D. virginiana* mothers in a recent study, vertical transmission has been suggested as a possible mechanism for amplifying infection among these marsupials [27]. However, if vertical transmission occurs in opossums, it remains unclear whether it happens during pregnancy or lactation.

The objective of the current study was to investigate the possible contribution of urban opossums to non-vectorial transmission of *T. cruzi* in the metropolitan area of Merida, Yucatan, in southeastern Mexico. We evaluated the presence of *T. cruzi* in blood and anal gland secretions from urban opossums, in the milk of opossum mothers breastfeeding their offspring, as well as in the blood of offspring from positive mothers. Additionally, we genotyped the parasite in all positive samples and quantified the parasite load of positive blood samples. The possible implications in non-vector transmission mechanisms are discussed.

## Methods

### Ethics statement

All methods of animal handling and sample collection were approved by the Institutional Animal Care and Use Committee of the Autonomous University of Yucatan (IACUC-03-2021). Opossums were obtained through an agreement with the Faculty of Veterinary Medicine and Zootechnics (FMVZ) of the Autonomous University of Yucatan (UADY) and the Secretariat

of Sustainable Development (SDS) of the municipality of Merida. The latter manages reports from citizens regarding opossums found in Merida's homes/neighborhoods for relocation.

## Study area and opossum capture methods

Opossums were sampled in urban settings from Merida, Uman and Conkal, within the metropolitan area of Merida, located in the northwestern region of the state of Yucatan (Southeastern Mexico), between March 2022 and March 2024 (Fig 1). According to the most recent data from 2020, the population of the metropolitan area of Merida exceeds 1.3 million inhabitants [28]. Animals were collected by homeowners in the vicinity of their homes in environments

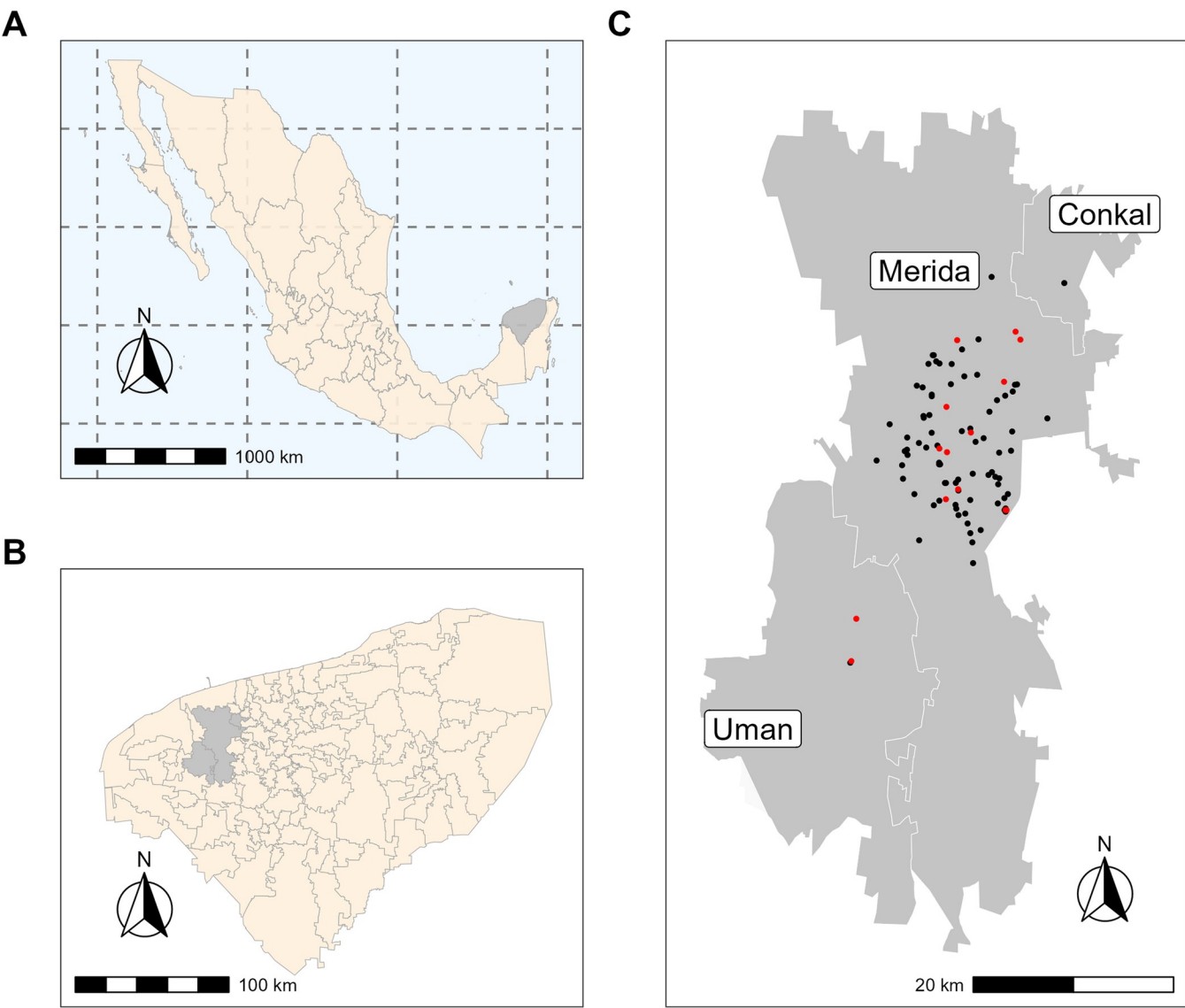

**Fig 1. Geographic distribution of opossum sampling sites.** (A) Map of Mexico, with the state of Yucatan (shaded area). (B) Map of the state of Yucatan. The gray area highlights the metropolitan area of Merida where opossum sampling was conducted. (C) Sampling sites within the metropolitan area of Merida. Red dots indicate urban setting locations where opossums tested positive for *T. cruzi*, while black dots indicate urban setting locations where opossums tested negative. The map was generated using RStudio (version 2023.06.1), integrating geolocation data collected during the study along with publicly accessible municipal boundary data for Yucatan, Mexico, obtained from INEGI (https://www.inegi.org.mx/app/biblioteca/ficha.html?upc=889463770541), and used in compliance with the "Terms of Free Use of INEGI Information" (https://www.inegi.org.mx/inegi/terminos.html).

such as backyards, parks and waste dumps and transported to the FMVZ of the UADY in cardboard boxes or pet carriers designed for dogs or cats. Additionally, other opossums were captured in the vicinity of some homes following reports to the SDS. In these cases, some animals were picked up manually by the research team, handled by grasping the tail with one hand and securing the head dorsally at the neck with the other before being placed in transport boxes, while others were captured using Tomahawk traps (66 x 23 x 23 cm, Tomahawk Live Trap Co.) baited with seasonal fruits (pineapple, sapota, or banana). Traps were set in the afternoon and checked early the following morning. Trap locations and capture sites were georeferenced. For opossums collected by homeowners, approximate location data were provided by them. However, sometimes, precise geolocation data was unavailable (S1 Table). A map of the sampling locations was generated using RStudio (version 2023.06.1), integrating geolocation data collected during the study along with publicly accessible municipal boundary data for Yucatan, Mexico, obtained from INEGI (https://www.inegi.org.mx/app/biblioteca/ficha.html?upc=889463770541) (Fig 1C). This information was used in compliance with the "Terms of Free Use of INEGI Information" (https://www.inegi.org.mx/inegi/terminos.html).

## Morphometric data recording and sample collection procedures

The opossums were initially immobilized by holding their tail with one hand while securing their neck with the other, positioning the fingers beneath the ears to control head movement. They were then placed in a lateral recumbent position, maintaining hold of the neck and positioning the hand holding the tail over the hip, applying gentle pressure to prevent rotation and thus ensure immobilization (Fig 2A).

Morphometric data, including weight, length, and sex of the opossums were recorded. In the case of females, their reproductive condition was determined by inspecting the marsupial pouch for embryos. Taxonomic identification was based on external characteristics such as the color of the hair on the cheeks, the length of the dark area of the tail, and the color of the secretion from the anal glands [19], but genetic confirmation of the species was also performed (see below). Age was determined based on total length, weight, and the sequence of teeth eruption [29]. Approximately 1 mL of blood was collected by venipuncture from the caudal vein using 1 mL syringes with 25G x 16 mm needles under aseptic conditions (Fig 2A). Additionally, between 0.5 and 1 mL of blood was collected from all offspring originating from the *T. cruzi*-positive mothers, when possible. Samples were deposited in tubes with EDTA. Similarly, samples of anal gland secretions, and of milk for lactating females, were collected through manual compression and placed in individual sterile tubes (Fig 2B and 2C). For the latter types of samples, volumes varying from 0.05 to 1 mL were obtained. After sampling, opossums were released in a semi-rural area in the southern outskirts of Merida, unless they presented a condition incompatible with relocation, such as *T. cruzi* infection or severe injuries jeopardizing their survival.

## DNA extraction

DNA was extracted from blood, anal gland secretions, and milk samples using the commercial DNeasy Blood & Tissue kit from QIAGEN following manufacturer's specifications. Purified DNA was quantified using a BioSpecnano spectrophotometer (Shimadzu Biotech, Kyoto, Japan). Consistent sample volumes were used for DNA extraction: 200 μL for blood samples, 100 μL for milk samples, and 25 mg for anal gland secretions.

## Genetic confirmation of opossum species

Taxonomic identification of opossum based on external characteristics was confirmed genetically using Sanger sequencing. Specific primers were designed to target the mitochondrial

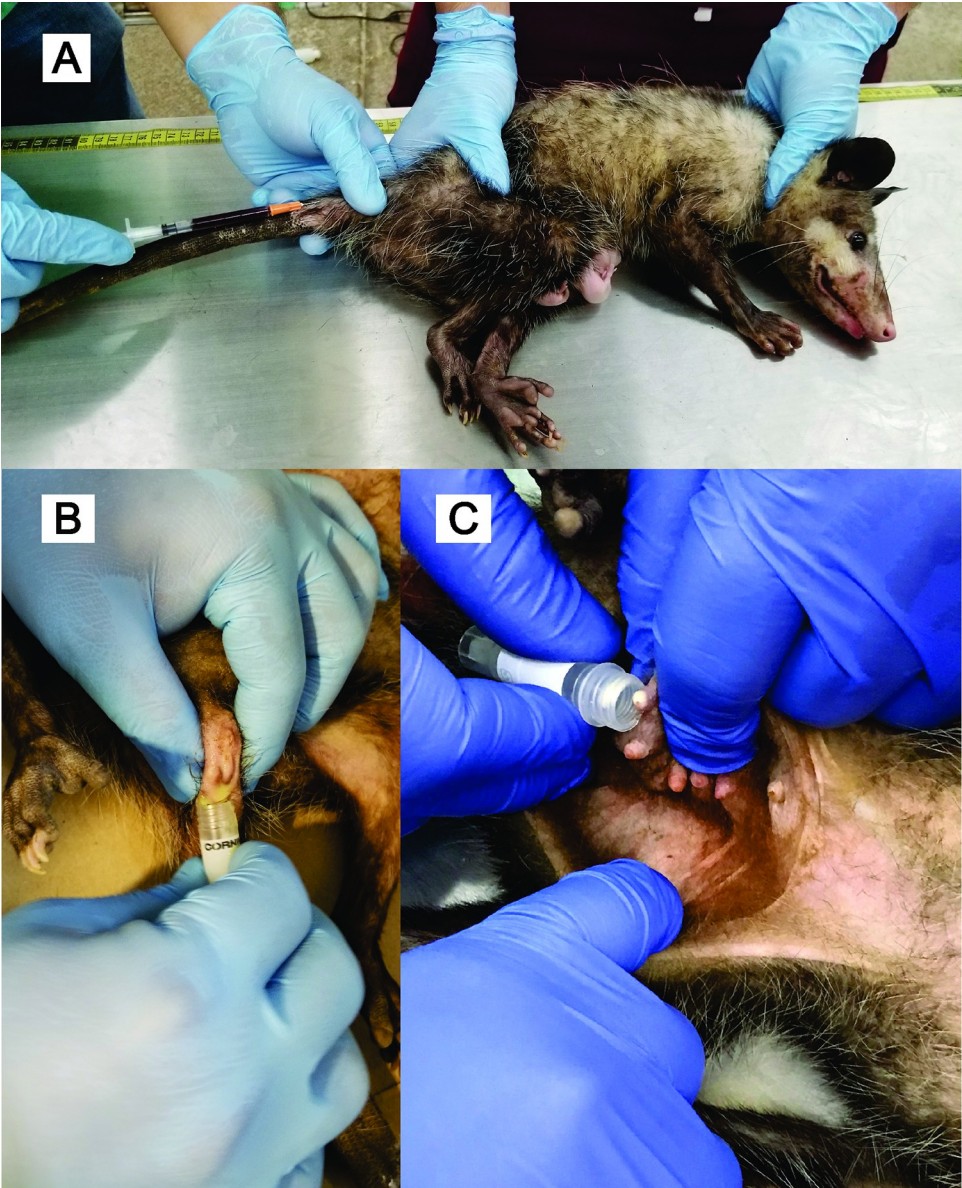

**Fig 2. Procedures for sample collection from opossums. (A) Blood sampling**: Blood was collected by venipuncture of the caudal vein. The image shows a *D. virginiana* female immobilized by physical restraint while blood sampling is performed. Several offspring are visible emerging from the marsupium. **(B) Anal gland secretion sampling**: Anal gland secretions were collected through manual compression. The image captures the posterior region of the opossum with the anal gland being compressed and the secretions collected in a tube. **(C) Milk sampling**: Milk was collected by gently expressing the mammary gland into a sterile collection tube. The image depicts the mammary gland being gently compressed to obtain the sample.

cytochrome b (Cytb) gene, after retrieving Cytb sequences of various opossum species in Genbank and aligning of these sequences using Mega 7 [30]: Cytb_did-F 5'-TTCGCAAAACA-CATCCACTCA-3' (forward primer) and Cytb_did-R 5'-GCCTGTTGGATTGYTTGATCC-3' (reverse primer). These primers allow for the amplification of a 620 pb fragment. PCR amplifications were performed using DNA extracted from opossum blood samples, in a final volume of 50 μL including 25 μL of 2X DreamTaq Green PCR Master Mix (Thermo Scientific), 0.2 μM of each primer, and 50–100 ng of DNA. PCR conditions were an initial denaturation at 94˚C

for 5 min, followed by 30 cycles of denaturation at 94°C for 30 s, annealing at 55°C for 30 s, and elongation at 72°C for 30 s, with a final cycle at 72°C for 7 min. To visualize the PCR products, 1.5% agarose gels stained with SYBR Safe (Invitrogen) were utilized. Purification and direct sequencing of both strands of PCR products were conducted by the company Eurofins Genomics in Konstanz, Germany. Sequences of both strands were aligned using CLUSTAL W [31] in BioEdit version 7.0.5.3 [32], and corrected in case of discrepancies through detailed analysis of the corresponding chromatograms. The corrected sequences were used for *Didelphis* species identification using BLAST. Additionally, the corrected sequences were aligned and trimmed, resulting in a 531 bp-long alignment which was analyzed using DnaSP version 6.12.03 [33] to identify the genetic diversity (haplotypes) of our population of opossum. Generated sequences were submitted to GenBank under accession numbers PQ516272-516278.

### Detection and quantification of parasite DNA in blood, anal gland secretions, and milk samples

The presence of *T. cruzi* DNA was assessed through PCR using the specific primers TCZ1 5'-CGAGCTCTTGCCCACACGGGTGCT-3' (forward primer) and TCZ2 5'-CCTCCAAGC AGCGGATAGTTCAGG-3' (reverse primer), which target a 188-bp sequence of *T. cruzi* nuclear DNA [34]. Visualization was performed using 1.5% agarose gels stained with SYBR Safe (Invitrogen). The parasite load in blood was quantified for each positive sample by quantitative real-time PCR (qPCR) using 2X KAPA SYBR FAST qPCR Master Mix and the primers described above. The parasite loads of each sample were calculated based on a standard curve and expressed as parasite DNA equivalent per mL of blood [35].

### *Trypanosoma cruzi* DTU determination

For a rapid typing of *T. cruzi* positive samples (blood, anal gland secretions or milk), DNA amplifications from positive samples were performed using a multiplex PCR with three forward primers targeting the intergenic region of the *T. cruzi* miniexon, namely Tc1: 5'-ACA CTTTCTGTGGCGCTGATCG-3', Tc2: 5'-TTGCTCGCACACTCGGCTGCAT-3', and Tc3: 5'-CCGCGWACAACCCCTMATAAAAATG-3' and a common reverse primer, namely Me: 5'-TACCAATATAGTACAGAAACTG-3'. The multiplex PCR allows the identification of three groups of DTUs based on specific molecular weights: TcI (200 bp), TcII/V/VI (250 bp), and TcIII/IV (150 bp) [36,37]. Visualization of the PCR products was performed using 1.5% agarose gels stained with SYBR Safe (Invitrogen).

### Statistical analysis

The overall prevalence was determined by dividing the total number of opossums testing PCR-positive on at least one sample type (blood, anal gland secretion or milk) by the total number of individuals sampled. Prevalence in milk samples was calculated by dividing the total number of opossums testing PCR-positive on this sample by the total number of female opossums with milk samples. A Wilson 95% confidence interval (CI$_{95\%}$) [38] was computed and presented for each prevalence value.

Chi-square tests were employed to analyze the association between the independent variables of sex, age, and Cytb haploype with the presence of *T. cruzi* DNA. A significance level of 5% was applied to all parameters tested ($p$-value $< 0.05$).

The normality of the parasite load data was assessed using the Shapiro-Wilk test. Given the non-normal distribution of the data, differences in parasite load between males and females, as well as between juveniles and adults, were evaluated using the Mann-Whitney U test. Additionally, the parasite load between Cytb haplotypes was analyzed using the Kruskal-Wallis test.

A significance level of 5% (*p*-value < 0.05) was considered for all tests, and Dunn's *post-hoc* tests were performed following significant Kruskal-Wallis tests. All statistical analyses were conducted using RStudio, version 2023.06.1.

## Results

A total of 102 opossums were sampled, comprising 82 adults and 20 juveniles, with a sex distribution of 39 males and 63 females (S1 Table). Among the females, 39 had offspring and 22 were lactating at the time of sampling. Peripheral blood and anal gland secretions samples were collected from each animal. Additionally, milk samples were obtained from lactating females, and blood samples were collected from 37 offspring of *T. cruzi*-positive mothers to investigate potential vertical transmission. All the opossums were collected in urban settings, 98 in the city of Merida, 3 in Uman, and 1 in Conkal (Fig 1C). Forty-nine were either captured or collected by homeowners, while the remaining 53 were collected by the research team. Of them, six were manually captured, and 47 were trapped using Tomahawk live traps.

All sampled opossums were identified as *D. virginiana* based on their physical characteristics. Sequencing of a 620 pb fragment of the opossum Cytb gene allowed obtaining, after alignment of the sequences of both strands and correction in case of any discrepancy by analyzing the corresponding chromatograms, clean sequences of up to 552 pb, which all presented a best identity with *D. virginiana* (range: 99.4–100% identity), confirming the species for all but one animal, for which sequences were not obtained. Among our population of opossums, seven Cytb haplotypes of *D. virginiana* were identified: CytbHap1 (n = 1), CytbHap2 (n = 11), CytbHap3 (n = 5), CytbHap4 (n = 53), CytbHap5 (n = 26), CytbHap6 (n = 2), and CytbHap7 (n = 5) (S1 Table and S1 File). Corresponding sequences were submitted to GenBank under accession numbers PQ516272-516278.

Out of the 102 opossums sampled, 16 (15.69%) tested positive for *T. cruzi* DNA in at least one sample (blood, anal gland secretion or milk) through PCR, including one female with positive samples in both blood and milk. The prevalence in blood was 14.71% (15/102) (Table 1). In anal gland secretion samples, the prevalence was 0.98% (1/102), with the only positive anal gland secretion sample originating from a female whose blood and milk samples tested negative. Only one milk sample (4.55%) of 22 tested was positive for *T. cruzi* DNA, and this sample came from an opossum that also tested positive in blood. Additionally, DTU determination was possible for all positive blood samples and showed that they belonged to TcI (S1 Table).

**Table 1. Prevalence of *T. cruzi* DNA in different samples obtained from male and female *D. virginiana*.**

|  | Sample type | Positive | Total samples | Prevalence (%) | Wilson CI$_{95\%}$ |
|---|---|---|---|---|---|
| **Males** | Blood | 7 | 39 | 17.95 | 8.95–32.56 |
|  | Anal gland secretions | 0 | 39 | 0 | 0–8.95 |
|  | **Overall*** | **7** | **39** | **17.95** | **8.95–32.56** |
| **Females** | Blood | 8 | 63 | 12.69 | 6.61–22.99 |
|  | Anal gland secretions | 1 | 63 | 1.58 | 0.28–8.48 |
|  | Milk | 1 | 22 | 4.55 | 0.83–22.07 |
|  | **Overall*** | **9** | **63** | **14.28** | **7.62–25.33** |
| **Total** | Blood | 15 | 102 | 14.71 | 9.08–22.91 |
|  | Anal gland secretions | 1 | 102 | 0.98 | 0.17–5.43 |
|  | Milk | 1 | 22 | 4.55 | 0.83–22.07 |
|  | **Overall*** | **16** | **102** | **15.69** | **9.74–24.23** |

*Number of positive opossums considering all sample types.

Kruskal-Wallis, $\chi^2(2) = 5.42$, $p = 0.067$, $n = 15$

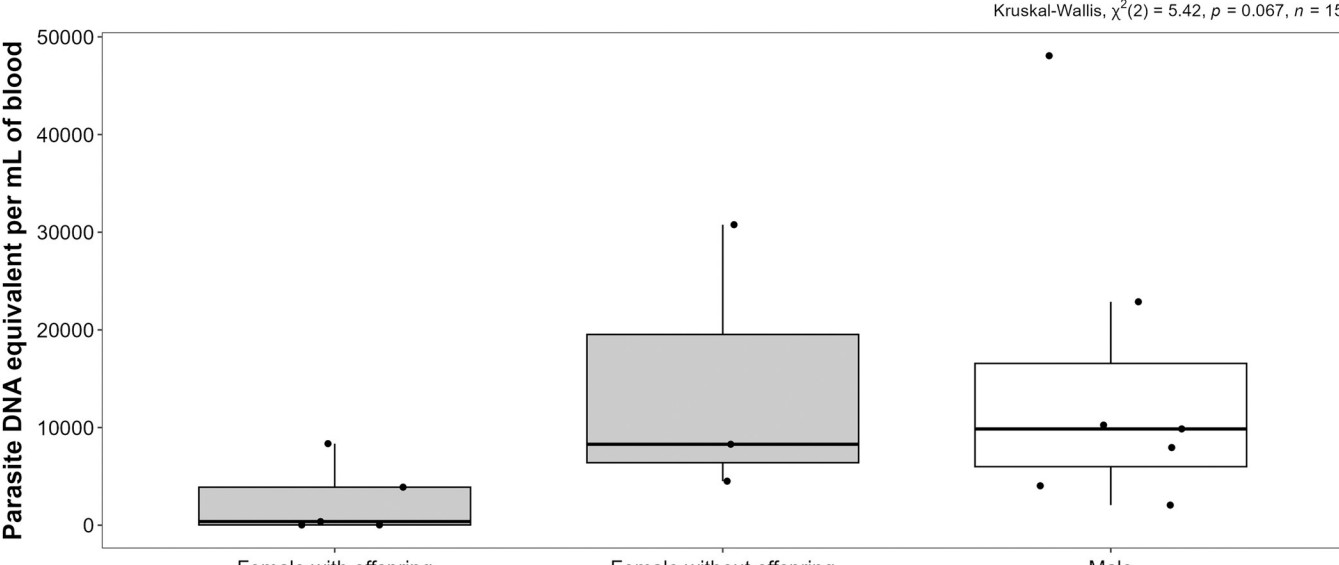

**Fig 3. Parasite load by sex and reproductive condition in *Didelphis virginiana*:** This boxplot illustrates the distribution of *T. cruzi* parasite loads (expressed as parasite DNA equivalent per mL of blood) across different groups: females with offspring, females without offspring, and males. Each individual parasite load of opossum is depicted by a dot. The boxes represent the interquartile range (IQR), with the line inside indicating the median. The whiskers extend to the minimum and maximum values. The boxes for females (both with and without offspring) are shaded gray, whereas the box for males is white. The females with offspring had the lowest median parasite load, followed by females without offspring, while males exhibited the highest median parasite load. No statistical difference was found using the Kruskal-Wallis test (*p*-value = 0.067).

Among males, the prevalence was 17.95% (7/39), while it was 14.28% (9/63) in females (Table 1).

In adults, the prevalence of *T. cruzi* DNA was 17.07% (14/82, Wilson $CI_{95\%}$: 9.66–26.98), compared to 10% (2/20, Wilson $CI_{95\%}$: 1.23–31.69) in juveniles. All 37 offspring of *T. cruzi*-positive mothers were PCR-negative for the parasite. Of the seven *D. virginiana* haplotypes, *T. cruzi* DNA was found in three, with the following prevalences: CytbHap2: 18.18% (2/11, Wilson $CI_{95\%}$: 2.28–51.78), CytbHap4: 17.31% (9/52, Wilson $CI_{95\%}$: 8.23–30.33), and CytbHap5: 20.83% (5/24, Wilson $CI_{95\%}$: 7.13–42.15).

None of the independent variables: age (*p*-value = 0.66), sex (*p*-value = 0.83), or haplotype (*p*-value = 0.81), were significantly associated with the presence of *T. cruzi* DNA using Chi-square tests.

When evaluating the parasite load in blood, no significant difference was found between juveniles and adults (Mann-Whitney U test, *p*-value = 0.571), nor between males and females (Mann-Whitney U test, *p*-value = 0.152). However, when analyzing the parasite load by sex and the reproductive condition of the females, an interesting pattern emerged, as females with offspring had the lowest parasite loads (2,536 ± 3,637 SD), followed by females without offspring (14,522 ± 14,198 SD), while males exhibited the highest parasite loads (15,015 ± 16,030 SD) (Fig 3). Nevertheless, this pattern did not reach statistical significance (Kruskal-Wallis test, *p*-value = 0.067). When comparing the parasite load among the three haplotypes, CytbHap2, CytbHap4, and CytbHap5, no statistical difference was found (Kruskal-Wallis test, *p*-value = 0.063).

## Discussion

In Mexico, transmission of *T. cruzi* to humans occurs in most of the territory, and it is estimated that up to 4 million people harbor the parasite in the country [39]. In the state of

Yucatan in southeastern Mexico, up to 62 000 people are estimated to be infected, making it one of the states with highest infection prevalence [39–41]. While vectorial transmission is the main transmission route in endemic areas, and mainly led by *T. dimidiata* in Yucatan [42,43], other transmission routes need to be explored, as they are part of the complex eco-epidemiology of this parasitic infection. This study aimed to assess the presence of *T. cruzi* in various tissues of opossums from the metropolitan area of Merida, Yucatan, Mexico, and explore potential implications for non-vectorial transmission routes, as the participation of opossums has been suspected in alternative transmission mechanisms [44], such as oral transmission through anal gland secretions [8,18,20,21,25,45] and recently hypothesized vertical transmission [27].

All the collected opossums in this study were identified as *D. virginiana* using taxonomic keys and Cytb sequencing, agreeing with previous studies where this species was the most abundant in urban areas in southeastern Mexico [11,12]. We found a *T. cruzi* prevalence of 15.69% (16/102) in sampled opossums, which was lower than the previous reports in rural areas of the same region [10,11]. However, this rate is consistent with findings from a recent study conducted in urban settings from Texas, where 15% of *D. virginiana* opossums tested positive for *T. cruzi* [24]. Together, these results emphasize the importance of not overlooking urban environments as settings at risk for *T. cruzi* transmission [46,47], particularly in cities of southeastern Mexico such as Merida where, moreover, *T. dimidiata* are commonly found in human dwellings with prevalence of infection around 50% [48]. Although positive PCR results in opossum samples do not evidence the presence of viable *T. cruzi* parasites, the detection of parasite DNA in 14.71% of opossum blood samples indicates a potentially parasitemic population that could infect vectors through blood feeding [24,49]. Nevertheless, in previous studies aiming at identifying *T. dimidiata* bloodmeals in Yucatan, opossums have not frequently been identified as blood meal sources [48,50–54]. This observation, together with the results of our study, suggests that for opossums, which have an omnivorous diet including insects, natural infection likely occurs through the oral route while eating infected triatomines, as previously proposed [15–17].

On the other hand, despite the low prevalence of *T. cruzi* in anal gland secretions (0.98%), this may still play a significant role in the spread of the parasite, as opossums use to expel the content of their anal glands in response to threats or stress, potentially contaminating their feces and the environment [9,18,19,27,45]. This contamination pathway could facilitate parasite transmission, even in urban areas, thus, where the risk of contact with human remains a concern. The efficacy of this transmission route between opossums and to other mammals has been confirmed with *D. marsupialis* and *D. albiventris* [20,21]. In the same way, in a recent study performed in Colombia, *D. marsupialis* has been proposed as a key player in oral transmission, potentially explaining five reported cases of acute Chagas disease in humans [45]. Although natural infection of opossum anal gland secretions with *T. cruzi* is rarely documented, in southern United States, prevalence rates of 12 and 18.8% in anal gland secretions of *D. virginiana* were recently reported, contrasting with our findings [24,27]. In South America, infection prevalence in the anal gland secretions of *D. albiventris* ranging from 0 to 5% have been reported [21,25], while a prevalence of 4.5% has been reported with *D. marsupialis* [26]. This variability implies that colonization of the anal glands by *T. cruzi* is likely multifactorial. Factors such as opossum species, geographic region, parasite strain, and host health status may influence this colonization, beyond parasitemia alone, which has been suggested as a critical factor for the spread of the parasite to the anal glands [22].

In this study, no infected offspring were identified from *T. cruzi*-positive mothers. This observation might be attributed to the parasitic load, as females with offspring tended to have lower parasite loads compared to females without offspring and males, although this difference

was not statistically significant. It is hypothesized that congenital transmission of *T. cruzi* is known to be associated with high parasitemia during gestation, among other factors [55,56], which may account for the absence of positive offspring in this study. Likewise, studies focusing on naturally infected dogs have shown that the vertical transmission rate of *T. cruzi* is less than 1% [57], indicating that offspring from infected mothers are rarely affected. If vertical transmission occurs in opossums, it is likely that the infection rate would similarly be low. On the other hand, it has been reported that infection with *T. cruzi* can induce reproductive issues such as abortions and embryonic resorption [56,58]. Consequently, the absence of offspring in females with the highest parasitic loads could also be a result of the infection and might explain the absence of congenital transmission observed in this work.

Torhorst *et al.*, (2022) [27] suggested a potential vertical transmission mechanism in *D. virginiana*. The detection of *T. cruzi* in the milk of a lactating female in the current study is a noteworthy finding that underscores the importance of further investigating vertical transmission in opossums and highlights the significance of considering milk as a potentially infectious biological fluid in these marsupials. Unfortunately, in the current study, the offspring of the lactating female which was found positive in the milk could not be evaluated for infection.

Furthermore, our study confirms the strong association between opossums and the *T. cruzi* DTU TcI, as mentioned by other authors [24,59–62]. All typed samples belonged to this DTU. This finding also corroborates previous reports indicating that TcI is the most frequently identified *T. cruzi* DTU in Mexico [5,61,62].

In conclusion, these findings emphasize the complexity of *T. cruzi* transmission mechanisms in opossums. The low prevalence in anal gland secretions, the detection in milk, and a tendency to sex-based differences suggest the need for additional research to comprehend infection dynamics and factors influencing parasite transmission in opossums. This knowledge is crucial for designing comprehensive Chagas disease control strategies, considering not only vectors, but also the direct participation of opossums, in *T. cruzi* transmission. Moreover, the discovery that females with offspring tended to have lower parasite load compared to females without offspring and males could be a factor influencing potential vertical transmission of the parasite within opossum populations. Finally, while our results underscore the potential risk represented by *D. virginiana* in non-vectorial transmission in urban areas of southeastern Mexico, the low prevalence observed in the opossum samples with potential role in non-vectorial transmission, together with the fact that an important part of the transmission of *T. cruzi* to *D. virginiana* in Yucatan likely occurs through feeding of infected triatomines by these marsupials, suggests that the possibility of a sustained cycle of transmission in total absence of triatomines is rather unlikely.

## Supporting information

**S1 Table. Information of the opossums included in this study, indicating sample ID, sex, weight, length, age, offspring number, species, Cytb haplotype, results of *T. cruzi* DNA detection, parasite load and DTU identification.**
(XLSX)

**S1 File. Mitochondrial Cytochrome b (Cytb) haplotype sequences of *Didelphis virginiana*.**
(DOCX)

## Acknowledgments

The authors would like to express their gratitude to Ph.D. Vladimir Cruz, Ph.D. Eduardo Gutiérrez, Ph.D. Alonso Panti, Landy Pech, Hazeem Contreras, Lilibeth Maldonado, and Ph.

D. Vincent Manzanilla. Special thanks to all the students of the Medical Teaching Unit of the FMVZ-UADY for their assistance in caring for the opossums and collecting samples, and to the entire Parasitology Laboratory team at CIR-UADY for their unwavering support.

## Author Contributions

**Conceptualization:** Pedro Pablo Martínez-Vega, Hugo Ruiz-Piña, Antonio Ortega-Pacheco, Etienne Waleckx.

**Data curation:** Pedro Pablo Martínez-Vega, Christian Teh-Poot.

**Formal analysis:** Pedro Pablo Martínez-Vega, Christian Teh-Poot, Etienne Waleckx.

**Investigation:** Pedro Pablo Martínez-Vega, Marian Rivera-Pérez, Gabrielle Pellegrin, Antoine Amblard-Rambert, Jorge Andrés Calderón-Quintal, Christian Barnabé, Antonio Ortega-Pacheco, Etienne Waleckx.

**Methodology:** Pedro Pablo Martínez-Vega, Christian Barnabé, Hugo Ruiz-Piña, Antonio Ortega-Pacheco, Etienne Waleckx.

**Resources:** Hugo Ruiz-Piña, Antonio Ortega-Pacheco, Etienne Waleckx.

**Supervision:** Hugo Ruiz-Piña, Antonio Ortega-Pacheco, Etienne Waleckx.

**Writing – original draft:** Pedro Pablo Martínez-Vega, Etienne Waleckx.

**Writing – review & editing:** Pedro Pablo Martínez-Vega, Marian Rivera-Pérez, Gabrielle Pellegrin, Antoine Amblard-Rambert, Jorge Andrés Calderón-Quintal, Christian Barnabé, Christian Teh-Poot, Hugo Ruiz-Piña, Antonio Ortega-Pacheco, Etienne Waleckx.

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
