## [Decision Letter · Decision Letter 0]

31 Jul 2024

Dear Dr Waleckx,

Thank you very much for submitting your manuscript "Presence of Trypanosoma cruzi (TcI) in different tissues of Didelphis virginiana from urban areas of southeastern Mexico: epidemiological relevance and implications for non-vector transmission routes" for consideration at PLOS Neglected Tropical Diseases. As with all papers reviewed by the journal, your manuscript was reviewed by members of the editorial board and by several independent reviewers. In light of the reviews (below this email), we would like to invite the resubmission of a significantly-revised version that takes into account the reviewers' comments. 

Reviewers overall appreciated the value of the work, but require additional clarity in the methods, a figure showing sampling sites, and tempering of some statements to avoid over-interpreting results that did not reach statistical significance or only applied to a few animals.

We cannot make any decision about publication until we have seen the revised manuscript and your response to the reviewers' comments. Your revised manuscript is also likely to be sent to reviewers for further evaluation.

Sincerely,

Laura-Isobel McCall

Section Editor

Amy Gilbert

Section Editor

Reviewers overall appreciated the value of the work, but require additional clarity in the methods, a figure showing sampling sites, and tempering of some statements to avoid over-interpreting results that did not reach statistical significance or only applied to a few animals.

Reviewer's Responses to Questions

**Key Review Criteria Required for Acceptance?**

**Methods**

-Are the objectives of the study clearly articulated with a clear testable hypothesis stated?

-Is the study design appropriate to address the stated objectives?

-Is the population clearly described and appropriate for the hypothesis being tested?

-Is the sample size sufficient to ensure adequate power to address the hypothesis being tested?

-Were correct statistical analysis used to support conclusions?

-Are there concerns about ethical or regulatory requirements being met?

Reviewer #1: - Methods of opossum collection need to be more detailed. Were animals placed under sedation with chemical restraint? What does manually mean? 

- Which portions of Merida were the opossums collected? I also suggest a map of the region and collection points as it relates to the urbanized setting. Were they collected in the city central or on the outskirts of Merida? A GIS map showing collection sites and possibly opossum collection density is also important. 

- I would also include how many were collected by homeowners, "manually" and tomahawk trap. 

- Were animals released after biospecimen collection?

- Were all animals alive at time of biospecimen collection? I ask this question as some animals were captured by homeowners. Did they put them in a cage? 

- Did authors measure the amount of anal gland secretion and milk collected? Was this consistent between animals? Low volumes could explain some discrepancies in detection.

Reviewer #2: Yes.

**Results**

-Does the analysis presented match the analysis plan?

-Are the results clearly and completely presented?

-Are the figures (Tables, Images) of sufficient quality for clarity?

Reviewer #1: In the results section I also recommend a table or figure showing T. cruzi infected opossum as it relates to the city of Merida. A figure is preferred. This gives the reader a visual aspect to the findings. T. cruzi is being transmitted within an urbanized setting in a city with close to 1 million people. This is an important finding. As mentioned by the authors, there is growing evidence of zoonotic transmission to other hosts and humans via the anal gland secretions. Albeit, in this sample the detection was low, it was detected and has been detected in other studies.

Reviewer #2: Yes

**Conclusions**

-Are the conclusions supported by the data presented?

-Are the limitations of analysis clearly described?

-Do the authors discuss how these data can be helpful to advance our understanding of the topic under study?

-Is public health relevance addressed?

Reviewer #1: In the Discussion section I would also suggest adding:

- I would suggest some discussion for blood meal analysis of T. dimidiata that have been collected in Merida and close to or near the study sites. It appears that most studies have shown bugs collected outside the city or close to the city? This author team has done a lot of this work but it would important to illustrate whether infected triatomines are found in the landscape of these collected opossums. There important potential discussion point here which could suggest that T. cruzi is being transmitted from opossum to opossum without the need for the vector. 

- I would mention this study in the Discussion: https://pubmed.ncbi.nlm.nih.gov/37956445/

- Authors from a region of Boyaca, Colombia, have suggested potential for D. marsupialis to serve as a source for oral transmission to explain 5 cases of acute Chagas. 

- Another question for the authors. Do community members eat opossum in anyway? I would mention whether this practice is present and could lead to another source of oral transmission with undercooked meat. In some cultural practices, people will drink raw blood for treatment of aliments. This practice is seen in Mexico with armadillo and other regions like Colombia. https://pubmed.ncbi.nlm.nih.gov/38251211/

Reviewer #2: Yes In some instances the importance of alternate transmission route is overstated given only single observation of infected milk.

**Editorial and Data Presentation Modifications?**

Reviewer #1: (No Response)

Reviewer #2: (No Response)

**Summary and General Comments**

Reviewer #1: Overall, the manuscript is well written and investigates the importance of the Virginia Opossum in the life cycle of T. cruzi in Merida, Mexico. The work is sound and thorough for research questions poised. I have some suggestions for the manuscript which are listed in the Methods, Results and Conclustions, reports. 

One Major revision requests: 

- I would suggest that the authors avoid using "southeastern" as a descriptor of the study region. Southeastern Mexico is a large portion of Mexico, includes the states of Oaxaca, Chiapas, Tabasco, Campeche, Quintana Roo, and the Yucatan. I recommend that the authors state in the title, "... from urban areas of Merida, Mexico..." I would also suggest that the author really focus on the fact this work was conducted in Merida and not associate all urbans areas of southeastern Mexico. This is a very important distinction as other urbans areas may have more or less T. cruzi infected D. virginiana in the landscape and until the research has been conducted, we will not know the true prevalence in other urban settings. 

 Minor revision requests: 

- Line 76 and 77, I think the authors meant to state, ...a complex zoonosis that is potentially fatal."

Reviewer #2: Overall:

The authors contribute a well-written, well-analyzed investigation of T. cruzi DNA in D. virginiana opossums in the urban setting of Merida, Mexico to provide contemporary data on infection prevalence in this synanthropic species and observations of parasite DNA in milk and anal glad secretion (1 animal each). The work bring awareness to the urban disease ecology of Chagas, an overlooked area as most studies are of rural/impoverished/ranch settings. Opportunistic routes were used for access to animals, including the nuisance animals removed from people’s homes (these were simply picked up manually with no trap, very impressive!); these are a good study population given their epidemiological relevance as they enter homes where people live. The study design, analysis, and interpretation are straight-forward and this adds to a couple recent papers from different geographic areas in the US to bring attention to this reservoir species. Figure 1 is a nice addition to show real photos of processing mammals from the field. I have a few comments for clarification and look forward to seeing this published paper in the future.

Major Comments:

• I feel the manuscript would be improved if the authors can clarify in the Introduction or Discussion what the speculated epidemiological (human health) or ecological significance is, to discern if opossums have vertical transmission of T. cruzi? In a region where there are abundant vectors and vectorial transmission, what is the importance to know if vertical transmission also occurs? Perhaps further presentation of the vector community/infection prevalence, and how human case reports compare to other regions of Mexico, would be useful. Given the presence of vectors in homes, I don’t believe it is being suggested that opossums are maintaining parasite in the absence of vectors.

• Although the overall sample size is over 100 opossums, the sample size of milk and blood from mothers is much lower. In some cases, it seems too much is speculated from the single observation of T. cruzi positive milk from infected mother. In the ‘Conclusions/Significance’ section of Abstract, it is stated that detection of T. cruzi in anal gland secretions (plural)….’ But only a single secretion tested positive in this study. 

• The statistical testing failed to detect a difference among parasite loads among the lactating vs. non-lactating females vs. males (with very large SD that was equal to or greater than the median value), yet the authors often refer to the ‘lower parasite load in females with offspring’ (e.g., abstract line 53). The Discussion (line 374) includes language about a lower parasite load in a study group. That statement is not supported statistically and so the presentation and discussion of these results should be corrected. 

Minor Comments:

• Line 30: Change ‘Opossums may have the ability to harbor…’ to ‘Opossums can harbor…’ as this has been shown for some species.

• Lines 76-77: Sentence seems to be missing a could words, perhaps insert ‘that is’ as follows: ‘… a complex zoonosis that is potentially fatal’.

• Re: the methods for animal processing Was no chemical immobilization used for the opossums? What was the fate of the animals after sample collection? Euthanasia or release details should be provided. 

• Did all the samples that tested positive using the TCZ1/2 PCR, also test positive in the assay to quantify parasite load; and did the all also test positive in the DTU determination assay? Knowing that each assay has a different sensitivity, some clarification on the congruence among assays in this sample set will be useful.

• If you have not already, can you deposit the study sequences (the opossum sequences showing the different haplotypes) into an open access sequence database?

• Lines 280-282 and Figure 2: Since quantification was based on comparison to standard curve of DNA, perhaps clarify that this is parasite equivalents of DNA rather than parasites.

• Line 369: change gender to sex, as sex is a biological concept, while gender is a social and cultural concept.

PLOS authors have the option to publish the peer review history of their article (what does this mean?). If published, this will include your full peer review and any attached files.

Reviewer #1: No

Reviewer #2: No
---

## [Decision Letter · Decision Letter 1]

26 Nov 2024

Dear Dr Waleckx,

We are pleased to inform you that your manuscript 'Presence of Trypanosoma cruzi (TcI) in different tissues of Didelphis virginiana from the metropolitan area of Merida, Southeastern Mexico: epidemiological relevance and implications for non-vector transmission routes' has been provisionally accepted for publication in PLOS Neglected Tropical Diseases.

Best regards,

Laura-Isobel McCall

Section Editor

Victoria Brookes

Section Editor

Shaden Kamhawi

co-Editor-in-Chief

Paul Brindley

co-Editor-in-Chief

Reviewer's Responses to Questions

**Key Review Criteria Required for Acceptance?**

**Methods**

-Are the objectives of the study clearly articulated with a clear testable hypothesis stated?

-Is the study design appropriate to address the stated objectives?

-Is the population clearly described and appropriate for the hypothesis being tested?

-Is the sample size sufficient to ensure adequate power to address the hypothesis being tested?

-Were correct statistical analysis used to support conclusions?

-Are there concerns about ethical or regulatory requirements being met?

Reviewer #1: Authors have addressed adequately the comments and suggestions from both Reviewers.

**Results**

-Does the analysis presented match the analysis plan?

-Are the results clearly and completely presented?

-Are the figures (Tables, Images) of sufficient quality for clarity?

Reviewer #1: Authors have addressed adequately the comments and suggestions from both Reviewers.

**Conclusions**

-Are the conclusions supported by the data presented?

-Are the limitations of analysis clearly described?

-Do the authors discuss how these data can be helpful to advance our understanding of the topic under study?

-Is public health relevance addressed?

Reviewer #1: Authors have addressed adequately the comments and suggestions from both Reviewers.

**Editorial and Data Presentation Modifications?**

Reviewer #1: (No Response)

**Summary and General Comments**

Reviewer #1: Authors have made edits and adjustments that improve the presentation of this important work. I have no other suggestions at this time. I recommend accepting for publication.

PLOS authors have the option to publish the peer review history of their article (what does this mean?). If published, this will include your full peer review and any attached files.

Reviewer #1: **Yes: **Norman L. Beatty, University of Florida College of Medicine, Gainesville, Florida, USA

---

## [Editor Report · Acceptance letter]

5 Dec 2024

Dear Dr Waleckx,

We are delighted to inform you that your manuscript, "Presence of Trypanosoma cruzi (TcI) in different tissues of Didelphis virginiana from the metropolitan area of Merida, Southeastern Mexico: epidemiological relevance and implications for non-vector transmission routes," has been formally accepted for publication in PLOS Neglected Tropical Diseases.

Best regards,

Shaden Kamhawi

co-Editor-in-Chief

Paul Brindley

co-Editor-in-Chief
